# Exact Inference with Latent Variables in an Arbitrary Domain

## Abstract

We analyze the necessary and sufficient conditions for exact inference of a latent model. In latent models, each entity is associated with a latent variable following some probability distribution. The challenging question we try to solve is: can we perform exact inference without observing the latent variables, even without knowing what the domain of the latent variables is? We show that exact inference can be achieved using a semidefinite programming (SDP) approach without knowing either the latent variables or their domain. Our analysis predicts the experimental correctness of SDP with high accuracy, showing the suitability of our focus on the Karush-Kuhn-Tucker (KKT) conditions and the spectrum of a properly defined matrix. Running on a laptop equivalent, our method can achieve exact inference in models with over 10000 entities efficiently. As a byproduct of our analysis, we also provide concentration inequalities with dependence on latent variables, both for bounded moment generating functions as well as for the spectra of matrices. To the best of our knowledge, these results are novel and could be useful for many other problems.

## 1 Introduction

Generative network models have become a powerful tool for researchers in various fields, including data mining, social sciences, and biology (Goldenberg et al., 2010; Fortunato, 2010). With the emergence of social media in the past decade, researchers are now exposed to millions of records of interaction generated on the Internet everyday. One can note that the generic structure and organization of social media resemble certain network models, for instance, the Erdos-Renyi model, the stochastic block model, the latent space model, the random dot product model (Goldenberg et al., 2010; Newman et al., 2002; Young & Scheinerman, 2007). The analogy comes from the fact that, in a social network each user can be modeled as an *entity*, and the interaction of users can be modeled as edges. One common assumption is that nodes belong to different groups. In social networks this can be users' political view, music genre preferences, or whether the user is a cat or dog person. Another common assumption, often referred to as *homophily* in prior literature, suggests that entities from the same group are more likely to be connected with each other than those from different groups (Goldenberg et al., 2010; Hoff, 2008; Krivitsky et al., 2009). The core task of *inference*, also known as graph partitioning, is to partition the nodes into groups based on the observed interaction information (Abbe, 2018; Ke & Honorio, 2018; Fortunato, 2010).

In this paper, we are particularly interested in the class of *latent models* beyond graphs, with latent variables in arbitrary domains. In a latent model, every entity belongs to one of $k$ groups. Every entity is associated with a latent variable in some arbitrary latent domain. It is natural to assume that for entities from the same group, their associated latent variables follow the same probability distribution. The latent model is equipped with a function to measure the homophily of two latent variables. Finally, two entities have some affinity score depending on their homophily in the latent domain. In other words, similar entities are more likely to have a higher affinity score. We want to highlight that, for the particular case of binary (i.e., $\{0,1\}$) affinity scores, the latent model is a random graph model. The challenging problem problem we try to solve is to infer the true group assignments *without* observing the the latent variables nor knowing the latent domain.

In the past decade, there have actually existed a large amount of literature on network models, and most focus on the class of *fully observed models*, for example, the Erdos-Renyi Model, and the Stochastic Block Model. These models are called "fully observed", because there are no latent variables, and edges are generated based on the agreement of entity labels. Some efficient algorithms have also been proposed for inference in these fully observed models (Abbe et al., 2016; Bandeira, 2018; Hajek et al., 2016; Chen & Xu, 2014). On the other hand, there is limited research on the class of latent models. Researchers have motivated various network models with latent variables, including the latent space model (Hoff et al., 2002), the exchangeable graph model (Goldenberg et al., 2010), the dot product model (Nickel, 2008), the uniform dot product model (Young & Scheinerman, 2007), and the extremal vertices model (Daudin et al., 2010). However to the best of our knowledge, no efficient polynomial time algorithms with formal guarantees have been proposed or analyzed for *exact inference* in latent models.

In this paper we address the problem of exact inference in latent models with arbitrary domains. More specifically, our goal is to correctly infer the group assignment of every entity in a latent model without observing the latent variables or the latent domain. We also propose a polynomial-time algorithm for exact inference in latent models using semidefinite programming (SDP). We want to highlight that many techniques used in the analysis of fully observed models do not directly apply to latent models. This is because in latent models, affinities are no longer statistically independent. As a result, latent models are more challenging to analyze than fully observed models, such as the stochastic block model.

While SDP has been heavily proposed for different machine learning problems, our goal in this paper is to study the optimality of SDP for our more challenging model. Our analysis focuses on Karush-Kuhn-Tucker (KKT) conditions and the spectrum of a carefully constructed primal-dual certificate. For convex problems including SDPs, the KKT conditions are sufficient and necessary for strong duality and optimality (Boyd & Vandenberghe, 2004). To the best of our knowledge, we are providing the first polynomial time method for a generally computationally hard problem with formal guarantees. In general, problems involving latent variables are computationally hard and nonconvex, for instance, learning restricted Boltzmann machines (Long & Servedio, 2010) or structural Support Vector Machines with latent variables (Yu & Joachims, 2009). We test the proposed method on both synthetic and real-world datasets. Running on a laptop equivalent, our method can achieve exact inference in models with over 10000 entities in less than 30 minutes, suggesting the computational efficiency of our approach. It is worth mentioning that theoretical computer science typically assumes arbitrary inputs ("worst-case" computationally hard), whereas we assume inputs are generated by a probabilistic generative model. Our results could be seen as "average-case" polynomial time: we provide exact inference conditions with respect to the model parameters $(p, q)$.

**Summary of our contributions.** We provide a series of novel results in this paper:

- We propose the definition of the latent model class, which is highly general and subsumes several latent models from prior literature (see Table 1).
- We provide the first polynomial time algorithm for a generally computationally hard problem with formal guarantees. We also analyze the sufficient conditions for exact inference in latent models using a semidefinite programming approach.
- For completeness, we provide an information-theoretic lower bound on exact inference, and we analyze when nonconvex maximum likelihood estimation is correct.
- As a byproduct of our analysis, we provide latent conditional independence (LCI) concentration inequalities, which are a key component in the analysis of latent models (see Remark 7). To the best of our knowledge, these results are novel and could be useful for many other latent model problems.

## 2 Preliminaries

In this section, we introduce the notations that will be used in later sections. First we provide the definition of the class of latent models.

**Definition 1** (Class of latent models)**.** A model $\mathcal{M}$ is called a *latent model* with $n$ entities and $k$ clusters, if $\mathcal{M}$ is equipped with structure $(\mathcal{X}, f, \mathcal{P})$ satisfying the following properties:

| Models | $\mathcal{X}$ | $f(x_i, x_j)$ |
|---|---|---|
| Latent space model | $\mathbb{R}^d$ | $\exp(-\|x_i - x_j\|^2)$ |
| Exchangeable graph model | $\{0,1\}^d$ | $\exp(-\|x_i - x_j\|_1)$ |
| Dot product graph (DPG) | $\mathbb{R}^d$ | $g(x_i \cdot x_j)$ |
| Uniform DPG | $[0,1]^d$ | $g(x_i \cdot x_j)$ |
| Extremal vertices model | $\{x \in \mathbb{R}_+^d \mid x_i \geq 0, \sum_{i=1}^d x_i = 1\}$ | $g(x_i \cdot x_j)$ |
| Kernel latent variable model | $\mathbb{R}^d$, sets, graphs, text, etc. | $g(K(x_i, x_j))$ |

Table 1: Comparison of various latent models, including the latent space model (Hoff et al., 2002), the exchangeable graph model (Goldenberg et al., 2010), the dot product graph (DPG) (Nickel, 2008), the uniform DPG (Young & Scheinerman, 2007), the extremal vertices model (Daudin et al., 2010), and the kernel latent variable model. In dot product models, $g : \mathbb{R} \rightarrow [0,1]$ is a function that normalizes dot products to the range of $[0,1]$. In kernel models, $K : \mathcal{X} \times \mathcal{X} \rightarrow \mathbb{R}$ is an arbitrary kernel function.

- $\mathcal{X}$ is an arbitrary latent domain;
- $f : \mathcal{X} \times \mathcal{X} \rightarrow [0,1]$ is a homophily function, such that $f(x, x') = f(x', x)$;
- $\mathcal{P} = (\mathcal{P}_1, \ldots, \mathcal{P}_k)$ is the collection of $k$ distributions with support on $\mathcal{X}$.

For simplicity we consider the balance case in this paper: each cluster has the same size $m := n/k$. Let $Z^* \in \{0,1\}^{n \times k}$ be the true cluster assignment matrix, such that $Z_{ij}^* = 1$ if entity $i$ is in cluster $j$, and $Z_{ij}^* = 0$ otherwise. For every entity $i$ in cluster $j$, nature randomly generates a latent vector $x_i \in \mathcal{X}$ from distribution $P_j$. A random observed affinity matrix $W \in [0,1]^{n \times n}$ is generated, such that the conditional expectation fulfills $\mathbb{E}_{W_{ij}} [W_{ij}|x_i, x_j] = f(x_i, x_j)$.

*Remark* 1. We use $[0,1]$ for $f$ and $W$ for clarity of exposition. Our results can be trivially extended to a general domain $[0, B]$ for $B > 0$ using the same techniques in later sections.

*Remark* 2. A particular case of the latent model is a random graph model, in which every entry $W_{ij}$ in the affinity matrix is binary (i.e., $W_{ij} \in \{0,1\}$) and generated from a Bernoulli distribution with parameter $f(x_i, x_j)$.

Our definition of latent models is highly general. In Table 1, we illustrate several latent models motivated from prior literature that can be subsumed under our model class by properly defining $\mathcal{X}$ and $f$.

In latent models, affinities are not independent if not conditioning on the latent variables. For example, suppose $i, j$ and $k$ are three entities. In fully observed models the affinities $W_{ij}$ and $W_{ik}$ are independent, but this is not true in latent models, as shown graphically in Figure 1. This motivates our following definition of *latent conditional independence (LCI)*.

**Definition 2** (Latent Conditional Independence). We say random variables $V = (v_1, v_2, \ldots)$ are *latently conditional independent* given $U$, if $v_1, v_2, \ldots$ are conditional independent given the unobserved latent random variable $U$.

## 2.1 Notations

We denote $[n] := \{1, 2, \ldots, n\}$. We use $\mathcal{S}_+^n$ to denote the $n$-dimensional positive semidefinite matrix cone, and $\mathbb{R}_+^n$ to denote the $n$-dimensional nonnegative orthant.

For simplicity of analysis, we use $z_i \in \{0,1\}^k$ to denote the $i$-th row of $Z$, and $z^{(i)} \in \{0,1\}^n$ to denote the $i$-th column of $Z$. We use $X = (x_1, \ldots, x_n)$ to denote the collection of latent variables.

Regarding eigenvalues of matrices, we use $\lambda_i(\cdot)$ to refer to the $i$-th smallest eigenvalue, and $\lambda_{\max}(\cdot)$ to refer to the maximum eigenvalue.

Regarding probabilities $\mathbb{P}_W \{\cdot\}, \mathbb{P}_X \{\cdot\}$, and $\mathbb{P}_{WX} \{\cdot\}$, the subscripts indicate the random variables. Regarding expectations $\mathbb{E}_W [\cdot], \mathbb{E}_X [\cdot]$, and $\mathbb{E}_{WX} [\cdot]$, the subscripts indicate which variables we are averaging over. We

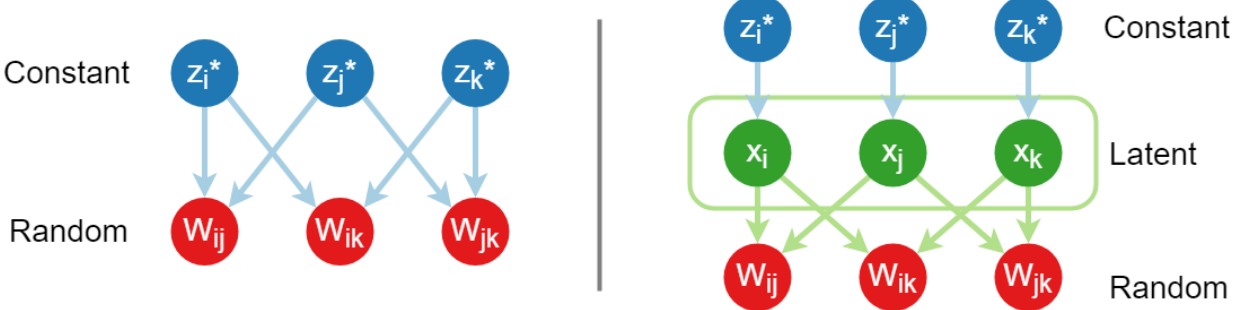

Figure 1: Comparison of fully observed models (left) and latent models (right). The blue nodes are the true (constant) labels $Z^*$. The green nodes are the unobserved latent random variables $X$ in our model. The red nodes are entries of the observed matrix $W$. In latent models, affinities are not independent without conditioning on latent variables. We say $W$'s are latently conditional independent given $X$.

use $\mathbb{P}_W \{\cdot \mid X\}$ to denote the conditional probability with respect to $W$ given $X$, and $\mathbb{E}_W [\cdot \mid X]$ to denote the conditional expectation with respect to $W$ given $X$.

For matrices, we use $\|\cdot\|$ to denote the spectral norm of a matrix, and $\|\cdot\|_F$ to denote the Frobenius norm. We use $\mathrm{tr}(\cdot)$ to denote the trace of a matrix, and $\mathrm{rank}(\cdot)$ to denote the rank. We use the notation $\mathrm{diag}(a_1, \ldots, a_n)$ to denote a diagonal matrix with diagonal entries $a_1, \ldots, a_n$. We also use $\mathbf{I}$ to refer to the identity matrix, and $1_n$ to refer to an all-one vector of length $n$. We use $\mathbb{S}^{n-1}$ to denote the unit $(n-1)$-sphere.

Let $S_i \in [n]^m$ denote the index set of the $i$-th cluster. For any vector $v \in \mathbb{R}^n$, we define $v_{S_i}$ to be the subvector of $v$ on indices $S_i$. Similarly for any matrix $V \in \mathbb{R}^{n \times n}$, we define $V_{S_i S_j}$ to be the submatrix of $V$ on indices $S_i \times S_j$. Denote the shorthand notation $V_{S_i} := V_{S_i S_i}$.

Define $d_i(S_l) := \sum_{j \in S_l} W_{ij}$ to be the degree of entity $i$ with respect to cluster $l$. Define shorthand notation $d_i$ to be the degree of entity $i$ with respect to its own cluster. Algebraically, we have $d_i := \sum_j W_{ij} z_i^{*\top} z_j^*$. We also denote $D := \mathrm{diag}(d_1, \ldots, d_n)$.

In the following sections we will frequently use the expected values related to the observed affinity matrix $W$. It would be tedious to derive every expression from $(\mathcal{X}, f, \mathcal{P})$. To simplify this, we introduce the following induced model parameters, which will be used throughout the paper.

**Definition 3** (Induced model parameters). In a latent model $\mathcal{M}$ equipped with structure $(\mathcal{X}, f, \mathcal{P})$, one can derive the following induced parameters: $p_l := \mathbb{E}_X [f(x_i, x_j) \mid i, j \in S_l]$ for $l \in [k]$, $q_{lr} := \mathbb{E}_X [f(x_i, x_j) \mid i \in S_l, j \in S_r]$ for $l, r \in [k], l \neq r$. Furthermore, one can define $p := \min_{l \in [k]} p_l$, $q := \max_{l,r \in [k], l \neq r} q_{lr}$. Note that both $p, q \in [0, 1]$.

## 3 Latent Conditional Independence Concentration Inequalities

In this section we provide new concentration inequalities with dependence on latent variables, both for bounded moment generating functions as well as for the spectra of matrices. It is worth highlighting that the notations in this section are not related to any particular model defined above. The proofs are in Appendix A.

**Lemma 1** (LCI tail bound). *Consider a finite sequence of random variables $V = (v_1, v_2, \ldots)$ that are LCI given $U$. Assume: 1) total expectation $\mathbb{E}_{v_i U} [v_i] = \mu_i$; 2) each term $v_i - \mathbb{E}_{v_i} [v_i \mid U]$ is sub-Gaussian with parameter $\sigma_i^2$ for all $U$; 3) sum of conditional expectations $\sum_i (\mathbb{E}_{v_i} [v_i \mid U] - \mu_i)$ is sub-Gaussian with parameter $\sigma_U^2$. Then for all positive $t$, $\mathbb{P}_{UV} \{\sum_i (v_i - \mu_i) \geq t\} \leq \exp\left(-\frac{t^2}{2(\sigma_U^2 + \sum_i \sigma_i^2)}\right)$.*

**Corollary 1** (LCI Hoeffding's inequality). *Consider a finite sequence of random variables $V = (v_1, v_2, \ldots)$ that are LCI given $U$. Assume: 1) total expectation $\mathbb{E}_{v_i U} [v_i] = \mu_i$; 2) bounded random variable $v_i \in [a_i, b_i]$*

*almost surely; 3) bounded sum of conditional expectations $\sum_i \mathbb{E}_{v_i}[v_i \mid U] \in [a_U, b_U]$ almost surely. Then for all positive $t$, $\mathbb{P}_{UV}\left\{\sum_i (v_i - \mu_i) \geq t\right\} \leq \exp\left(-\frac{2t^2}{(b_U - a_U)^2 + \sum_i (b_i - a_i)^2}\right)$.*

**Corollary 2** (LCI Bernstein inequality). *Consider a finite sequence of random variables $V = (v_1, v_2, \dots)$ that are LCI given $U$. Assume: 1) zero total expectation $\mathbb{E}_{v_i U}[v_i] = 0$; 2) bounded random variable $|v_i| \leq R$ almost surely; 3) bounded variance $\mathbb{E}_{v_i}\left[(v_i - \mathbb{E}_{v_i}[v_i \mid U])^2 \mid U\right] \leq \nu_i^2$ for all $U$, $\mathrm{Var}_U\left[\sum_i \mathbb{E}_{v_i}[v_i \mid U]\right] \leq \nu_U^2$. Then for all positive $t$, $\mathbb{P}_{UV}\left\{\sum_i v_i \geq t\right\} \leq \exp\left(-\frac{t^2/2}{\nu_U^2 + \sum_i \nu_i^2 + Rt/3}\right)$.*

**Lemma 2** (LCI matrix tail bound). *Consider a finite sequence of random symmetric matrices $V = (V_1, V_2, \dots)$ of dimension $d$ that are LCI given $U$. Let $M_i(U) := \mathbb{E}_{V_i}[V_i \mid U]$ be the conditional expectation of $V_i$ given $U$. Let $g$ be a function $g : (0, \infty) \to [0, \infty]$. Assume: 1) zero total expectation $\mathbb{E}_{V_i U}[V_i] = 0$; 2) there exists a sequence of symmetric matrices $\{A_i\}$ such that $\mathbb{E}_{V_i}\left[e^{\theta(V_i - M_i(U))} \mid U\right] \preceq e^{g(\theta) \cdot A_i}$ for $\theta > 0$ and for all $U$; 3) there exists a symmetric matrix $A_U$ such that $\mathbb{E}_U\left[e^{\theta \sum_i M_i(U)}\right] \preceq e^{g(\theta) \cdot A_U}$ for $\theta > 0$. Define the scale parameter $\rho := \lambda_{\max}\left(A_U + \sum_i A_i\right)$. Then for all positive $t$, $\mathbb{P}_{UV}\left\{\lambda_{\max}\left(\sum_i V_i\right) \geq t\right\} \leq d \cdot \inf_{\theta > 0} e^{-\theta t + g(\theta) \cdot \rho}$.*

**Corollary 3** (LCI matrix Bernstein inequality). *Consider a finite sequence of random symmetric matrices $V = (V_1, V_2, \dots)$ of dimension $d$ that are LCI given $U$. Let $M_i(U) := \mathbb{E}_{V_i}[V_i \mid U]$ be the conditional expectation of $V_i$ given $U$. Assume: 1) zero total expectation $\mathbb{E}_{V_i U}[V_i] = 0$; 2) bounded eigenvalue $\lambda_{\max}(V_i) \leq R$ almost surely; 3) bounded variance $\left\|\mathbb{E}_U\left[(\sum_i M_i(U))^2\right] + \sum_i \mathbb{E}_{V_i}\left[(V_i - M_i(U))^2 \mid U\right]\right\| \leq \sigma^2$ for all $U$. Then for all positive $t$, $\mathbb{P}_{UV}\left\{\lambda_{\max}\left(\sum_i V_i\right) \geq t\right\} \leq d \cdot \exp\left(-\frac{t^2/2}{\sigma^2 + Rt/3}\right)$.*

# 4 Polynomial-Time Regime with Semidefinite Programming

In this section we investigate the sufficient conditions for exactly inferring the group assignment of entities in latent models. An algorithm achieves *exact inference* if the recovered group assignment matrix $Z \in \{0,1\}^{n \times k}$ is identical to the true assignment matrix $Z^*$ up to permutation of its columns (without prior knowledge it is impossible to infer the order of groups).

**Overview of the proof.** Our proof starts by looking at a *maximum likelihood estimation (MLE)* problem (1), which cannot be solved efficiently (for more details see Section 4). We relax the MLE problem (1) to problem (2) (matrix-form relaxation), then to problem (3) (convex SDP relaxation). We ask under what conditions the relaxation holds (i.e., returns the groundtruth). Our analysis proves that, if the statistical conditions in Theorem 1 are satisfied, by solving the relaxed convex optimization problem (3), one can recover the true group assignment $Z^*$ perfectly and efficiently with probability tending to 1.

Our analysis can be broken down into two parts. In the first part we demonstrate that the exact inference problem in latent models can be relaxed to a semidefinite programming problem. It is well-known that SDP problems can be solved efficiently (Boyd & Vandenberghe, 2004). We employ *Karush-Kuhn-Tucker (KKT) conditions* in our proof to construct a pair of primal-dual certificates, which shows that the SDP relaxation leads to the optimal solution under certain deterministic spectrum conditions. In the second part we analyze the statistical conditions for exact inference to succeed with high probability. The proofs are in Appendix B.

## 4.1 SDP Relaxation

We first consider a maximum likelihood estimation approach to recover the true assignment $Z^*$. The use of MLE in graph partitioning and community detection literature is customary (Bandeira, 2018; Abbe et al., 2016; Chen & Xu, 2014). The motivation is to find cluster assignments, such that the number of edges within clusters is maximized. Recall that $z_i \in \{0,1\}^k$ is the $i$-th row of $Z$, and $z^{(i)} \in \{0,1\}^n$ is the $i$-th column of $Z$. Given the observed matrix $W$, the goal is to find a binary assignment matrix $Z$, such that $\sum_{i,j} W_{ij} z_i^\top z_j$ is maximized. In the matrix form, MLE can be cast as the following optimization problem:

$$
\begin{aligned}
\underset{Z}{\text{maximize}} \quad & \langle W, ZZ^\top \rangle \\
\text{subject to} \quad & Z \in \{0,1\}^{n \times k}, Z^\top 1_n = m 1_k, Z 1_k = 1_n.
\end{aligned}
\tag{1}
$$

where the last two constraints enforce that each entity is in one of the $k$ groups, and each group has size $m = n/k$.

Problem (1) is nonconvex and hard to solve because of the $\{0,1\}$ constraint. In fact, in the case of two clusters ($k = 2$) and 0-1 weights, the MLE formulation reduces to the Minimum Bisection problem, which is known to be NP-hard (Garey et al., 1976). To relax it, we introduce the cluster matrix $Y = ZZ^\top$. One can see that $Y$ is a rank-$k$, $\{0,1\}$ positive semidefinite matrix. Each entry is 1 if and only if the corresponding two entities are in the same group ($z_i = z_j$). Similarly we can define $Y^* = Z^*Z^{*\top}$ for the true cluster matrix. Then the optimization problem becomes

$$
\begin{aligned}
\underset{Y}{\text{maximize}} \quad & \langle W, Y \rangle \\
\text{subject to} \quad & Y_{ii} = 1\,, \forall i \in [n], Y1_n = m1_n, 1_n^\top Y = m1_n^\top \\
& Y \succeq_{\mathcal{S}_+^n} 0, Y \succeq_{\mathbb{R}_+^n} 0, \text{rank}\,(Y) = k\,.
\end{aligned}
\tag{2}
$$

Problem (2) is still nonconvex because of the rank constraint. By dropping this constraint, we obtain the main SDP problem:

$$
\begin{aligned}
\underset{Y}{\text{maximize}} \quad & \langle W, Y \rangle \\
\text{subject to} \quad & Y_{ii} = 1, \forall i \in [n], Y1_n = m1_n, 1_n^\top Y = m1_n^\top \\
& Y \succeq_{\mathcal{S}_+^n} 0, Y \succeq_{\mathbb{R}_+^n} 0\,.
\end{aligned}
\tag{3}
$$

Problem (3) is now convex and can be solved efficiently. A natural question is: under what circumstances the optimal solution to (3) will match the solution to the original problem (1)? To answer the question, we take a primal-dual approach. One can easily see there exists a strictly feasible $Y$ for the constraints in (3). Thus Slater's condition guarantees *strong duality* (Boyd & Vandenberghe, 2004). We now proceed to derive the dual problem.

**Lemma 3** (Dual problem)**.** *The dual of* (3) *is*

$$
\begin{aligned}
\underset{h, A, \Gamma}{\text{minimize}} \quad & \text{tr}\,(A) + 2mh^\top 1_n \\
\text{subject to} \quad & A - W + h1_n^\top + 1_n h^\top - \Gamma \succeq_{\mathcal{S}_+^n} 0 \\
& A \text{ is diagonal}, \Gamma_{S_i} = 0\,, \forall i \in [k], \Gamma \succeq_{\mathbb{R}_+^n} 0\,.
\end{aligned}
\tag{4}
$$

We now construct the primal-dual certificates to close the duality gap between problem (3) and (4).

**Lemma 4** (Primal-dual certificates)**.** *Let $P := \mathbf{I} - \frac{1}{m}1_m1_m^\top$ to be the projection onto the orthogonal complement of* span $(1_m)$. *By setting the dual variables as $h = \frac{\phi}{2}1_n, A = D - m\phi\mathbf{I}, \Gamma_{S_i} = 0, \forall i \in [k], \Gamma_{S_iS_j} = \phi 1_m1_m^\top + PW_{S_iS_j}P - W_{S_iS_j}, \forall i \neq j$, where $\phi \in \mathbb{R}$ is a constant to be determined later, the duality gap between* (3) *and* (4) *is closed.*

It remains to verify feasibility of the dual constraints in (4). It is trivial to verify that $A = D - m\phi\mathbf{I}$ is diagonal, and $\Gamma_{S_i} \succeq_{\mathbb{R}_+^m} 0$. We now summarize the dual feasibility conditions.

**Lemma 5** (Dual feasibility)**.** *Let $h, A, \Gamma$ be defined as in Lemma 4. If*

$$
\Lambda := D - m\phi\mathbf{I} - W + \phi 1_n1_n^\top - \Gamma \succeq_{\mathcal{S}_+^n} 0\,,
\tag{5}
$$

*and*

$$
\Gamma_{S_iS_j} \succeq_{\mathbb{R}_+^m} 0
\tag{6}
$$

*for every $i, j \in [k]$ with $i \neq j$, then the dual constraints in* (4) *are satisfied.*

**We also require the optimal solution to be unique.** This means $Y^* = Z^*Z^{*\top}$ should be the only optimal solution to problem (3). To do so we look into the eigenvalues of $\Lambda$ defined in Lemma 5. It is easy

to verify that every $z^{*(i)}$ is an eigenvector of $\Lambda$ with $\Lambda z^{*(i)} = 0$. To ensure uniqueness, it is sufficient to require that all other $n - k$ eigenvalues of $\Lambda$ are strictly positive. We now provide the following lemma about uniqueness.

**Lemma 6** (Uniqueness). *The convex relaxed problem* (3) *achieves exact inference and outputs the unique optimal solution $Y = Y^* = Z^* Z^{*\top}$, if*

$$\lambda_{k+1}(\Lambda) > 0 . \tag{7}$$

*Remark* 3. Why is the requirement of uniqueness reasonable? Because our latent models are generative, i.e., the ground truth $Z^*$ is unique and generates everything, including the latent variables $X$ and the observed matrix $W$ (see Figure 1). From the perspective of optimization, in some cases there may exist multiple optimal solutions, but we are only interested in the cases in which the preexisting groundtruth $Z^*$ is returned. In fact, the requirement of uniqueness is customary in generative models (Abbe et al., 2016; Bandeira, 2018; Chen & Xu, 2014).

Combining the results above, we now give the sufficient conditions for exact inference.

**Lemma 7** (Deterministic sufficient conditions). *Let $h, A, \Gamma$ be defined as in Lemma 4. If*

$$\Gamma_{S_i S_j} \succeq_{\mathbb{R}_+^m} 0 \tag{8}$$

*for every $i, j \in [k]$ with $i \neq j$, and*

$$\lambda_{k+1}(\Lambda) > 0 , \tag{9}$$

*then $Y^* = Z^* Z^{*\top}$ is the unique primal optimal solution to* (3)*, and $(h, A, \Gamma)$ is the dual optimal solution to* (4)*.*

Note that Lemma 7 gives the deterministic condition for our SDP relaxation to succeed. In the following two sections, we characterize the statistical conditions for (8) and (9) to hold with probability tending to 1.

### 4.2 Entrywise Nonnegativity of $\Gamma$

In this section we analyze the statistical conditions for (8) to hold with high probability. From Lemma 4 it follows that $\Gamma_{S_i S_j} = \phi 1_m 1_m^\top + P W_{S_i S_j} P - W_{S_i S_j}, \forall i \neq j$. To ensure dual feasibility, it is necessary to ensure that every entry in $\Gamma_{S_i S_j}$ is nonnegative with high probability by setting a proper $\phi$.

We now present the condition for (8) to hold with high probability.

**Lemma 8** (Choice of $\phi$). *If $\phi \geq q + O\left(\sqrt{\frac{k \log n}{n}}\right)$, then $\Gamma_{S_i S_j} \succeq_{\mathbb{R}_+^m} 0$ holds for every $i, j \in [k]$ with probability at least $1 - O\left(\frac{1}{n}\right)$.*

*Remark* 4. To ensure nonnegativity, one may think about setting $\phi$ to be some sufficiently large constant (for example, set $\phi = 2$). This is not going to work, however, as the choice of $\phi$ also plays a critical role in the analysis of (9) in the next section. In order to obtain a tighter final result, it is necessary to pick the smallest possible $\phi$, without breaking the nonnegativity of $\Gamma$. For further details see Lemma 9.

### 4.3 Statistical Conditions of Efficient Inference

In this section we analyze the statistical conditions for (9) to hold with high probability. To do so, we first look at the expectation of $\Lambda$.

**Lemma 9** (Eigenvalue of expectation). *It follows that*

$$\lambda_{k+1}\left(\mathbb{E}_{WX}[\Lambda]\right) \geq m(p - \phi) . \tag{10}$$

*Remark* 5. The expectation above shows why the choice of $\phi$ matters. With a larger $\phi$, one has less degree of freedom to work with, in terms of the concentration inequalities.

The next step is to show that the eigenvalue of $\Lambda$ will not deviate too much from its expectation, so that $\lambda_{k+1}(\Lambda)$ is greater than 0 with high probability. In fact we have the following lemma.

**Lemma 10** (Sufficient concentration conditions)**.** *Assuming that $\phi < p$. To prove* (9) *holds with high probability, it is sufficient to prove*

$$\min_i(d_i - \mathbb{E}_{WX}[d_i]) + \frac{m}{2}(p - \phi) > 0 \tag{11}$$

*and*

$$-\lambda_{\max}(W - \mathbb{E}_{WX}[W]) + \frac{m}{2}(p - \phi) > 0 \tag{12}$$

*hold with high probability.*

We now present the statistical conditions for exact inference of latent models using semidefinite programming.

**Theorem 1.** *In a latent model of $k$ clusters and $n$ entities, with induced parameters $(p, q)$ as in Definition 3, if $\sum_{j=1}^{n}(\mathbb{E}_W[W_{ij} \mid X] - \mathbb{E}_{WX}[W_{ij}])$ is sub-Gaussian with parameter $O(n)$ for all $i \in [n]$, and $\left\|\mathbb{E}_X\left[(\mathbb{E}_W[W \mid X] - \mathbb{E}_{WX}[W])^2\right]\right\| = O(n)$, then the SDP-relaxed problem* (3) *achieves exact inference, i.e., $Y = Y^* = Z^*Z^{*\top}$, with probability at least $1 - O\left(\frac{1}{n}\right)$, as long as*

$$\frac{(p-q)^2}{k^2} = \Omega\left(\frac{\log n}{n}\right). \tag{13}$$

*Remark* 6. Our theorem requires that the deviation between the conditional expectations $\mathbb{E}_W[W_{ij} \mid X] = f(x_i, x_j)$ and the total expectations $\mathbb{E}_{WX}[W_{ij}] = p$ (or $q$), to be bounded above by some variance. Similar deviation bounds are necessary for latent models in general. To illustrate this, in Appendix C we discuss a counterexample, which fulfills the final condition above but not the deviation assumptions.

*Remark* 7. The LCI inequalities are a key component in our analysis. This is because in our model, $W$ depends on $\mathbb{E}_W[W \mid X]$, and $\mathbb{E}_W[W \mid X]$ depends on $\mathbb{E}_{WX}[W]$. Regular matrix concentration requires two steps: concentrating the observed random matrix $W$ around the conditional expectation $\mathbb{E}_W[W \mid X]$ using matrix Bernstein inequality, then concentrating $\mathbb{E}_W[W \mid X]$ around the total expectation $\mathbb{E}_{WX}[W]$ using matrix Chebyshev inequality. This approach gives the final bound in the form of $(p - q)^2/k^2 = \Omega(n^2)$, which is much worse than the $\Omega(\log n/n)$ rate. On the other hand, the LCI inequalities can be applied in one single step in the analysis, while giving us a tighter bound.

## 4.4 Reparametrization of Latent Space Model

As an example, in this section we present the latent space model (Hoff et al., 2002). We show the latent space model can be subsumed by our latent model class as in Definition 1, and Theorem 1 provides the statistical condition for exact inference. The proofs can be found in Appendix D.

**Definition 4** (Symmetric latent space model)**.** Let $n$ be a positive even integer, $\sigma > 0$, $d \in \mathbb{Z}^+$, and $\mu \in \mathbb{R}^d$, $\mu \neq 0$. In a symmetric latent space model with $n$ nodes and two clusters $\{+1, -1\}$, nature picks the groundtruth $y^*$ randomly from the space $\mathcal{Y} = \{y : y \in \{\pm 1\}^n, 1^\top y = 0\}$. Each latent vector $x_i$ is generated from the $d$-dimensional Gaussian distribution $\mathcal{N}_d(y_i^*\mu, \sigma^2 I)$. A random graph $\mathcal{G}$ is generated from $X = (x_1, \ldots, x_n)$, such that for every pair of nodes $i$ and $j$, $(i, j)$ is an edge of $\mathcal{G}$ with probability $\exp(-\|x_i - x_j\|^2)$.

**Corollary 4** (Exact inference in latent space model)**.** *In a $d$-dimensional symmetric latent space model of $n$ entities, with parameters $(\mu, \sigma)$ as in Definition 4, if the order of $d = \Theta(\log n)$ and $\mu, \sigma$ are constant, then the SDP-relaxed problem* (3) *achieves exact inference with probability at least $1 - O\left(\frac{1}{n}\right)$, as long as*

$$(4\sigma^2 + 1)^{-d}\left(1 - \exp\left(-\frac{4\|\mu\|^2}{4\sigma^2 + 1}\right)\right)^2 = \Omega\left(\frac{\log n}{n}\right).$$

## 5   Additional Analysis

In this section, for completeness, we also provide an information-theoretic lower bound on exact inference (i.e., the impossible regime), and we analyze when (nonconvex) maximum likelihood estimation is correct (i.e., the hard regime). The proofs are in Appendix E.

### 5.1   Impossible Regime

In this section we analyze the necessary conditions for exact inference of latent models. Our goal is to characterize the information-theoretic lower limit of any algorithm for inferring the true labels $Z^*$ in our model. More specifically, we would like to infer labels $\hat{Z}$ given the observation of the adjacency matrix $W$. Also note that we do not observe the collection of latent variables $X$. We present the following information-theoretic lower bound for our model.

**Claim 1.** Let $Z^*$ be the true assignment matrix sampled uniformly at random. In a latent model of $k$ clusters and $n$ entities, with induced parameters $(p, q)$ as in Definition 3, if

$$\frac{p}{k}\log(p/q) = O\left(\frac{1}{n}\right),$$

then the probability of error $\mathbb{P}\left\{\hat{Z} \neq Z^*\right\} \geq 1/2$, for any algorithm that a learner could use for picking $\hat{Z}$.

### 5.2   Hard Regime with Maximum Likelihood Estimation

In this section we analyze the conditions for exact inference of the true labels in latent models using nonconvex maximum likelihood estimation by solving optimization problem (1). We call this the hard regime because without some convex relaxation, enumerating $Z$ takes $O(k^n)$ iterations. The problem can be rewritten in the following square matrix form:

$$\underset{Y \in \mathcal{Y}}{\text{maximize}} \quad \langle W, Y \rangle, \tag{14}$$

where

$$\mathcal{Y} = \{ZZ^\top \mid Z \in \{0, 1\}^{n \times k}, Z^\top 1_n = \frac{n}{k}1_k, Z1_k = 1_n\},$$

is the space of all feasible solutions. We now state the conditions for exact inference of latent models using maximum likelihood estimation.

**Claim 2.** In a latent model of $k$ clusters and $n$ entities, with induced parameters $(p, q)$ as in Definition 3, if $\sum_{i,j}(\mathbb{E}_W[W_{ij} \mid X] - \mathbb{E}_{WX}[W_{ij}])$ is sub-Gaussian with parameter $O(n)$ for all $Y$, then the maximum likelihood estimation (14) achieves exact inference, i.e., $Y = Y^* = Z^* Z^{*\top}$, with probability at least $1 - O\left(\frac{1}{n}\right)$, as long as

$$\frac{(p-q)^2}{k} = \Omega\left(\frac{\log n}{n}\right).$$

## 6   Experimental Validation

In this section, we validate the proposed program (3) and Theorem 1 through synthetic experiments.

**Experiment 1**: We validate the proposed method on a latent space model using CVX (Grant & Boyd, 2014). We pick $\mathcal{X} = \mathbb{R}^2$ as the latent domain. We generate the latent variables using Gaussian distributions, such that $P_1 = \mathcal{N}_2(\mu, \sigma^2 \mathbf{I})$, $P_2 = \mathcal{N}_2(-\mu, \sigma^2 \mathbf{I})$, where $\mathcal{N}$ denotes the Gaussian distribution. We set the homophily function $f(x, x') = \exp(-\|x - x'\|^2)$, and each entry $W_{ij}$ is sampled from Bernoulli distribution with parameter $f(x_i, x_j)$. The parameters in our simulations is $\|\mu\|$. We iterate $\|\mu\|$ from 0.1 to 20 with an interval of 0.1. For each value of $\|\mu\|$, we run 20 trials and calculate the empirical probability that CVX returns the correct cluster matrix: $\mathbb{P}\{Y = Y^*\}$.

We plot the empirical probability of exact inference against value $C$, which is defined as $C := n(p-q)^2/\log n$ in the x-axis. Note that $C$ is equal to the left-hand side of (13) divided by its right-hand side, and $p$, $q$ are the induced parameters calculated as in Definition 3. Our result suggests that as $C$ gets larger, the proposed algorithm achieves exact inference with high probability tending to 1. This matches our theoretic findings in Theorem 1.

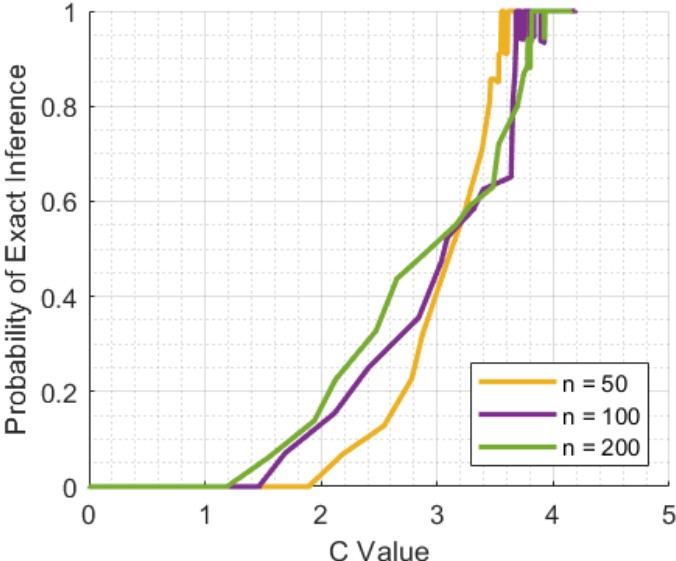

Figure 2: Simulations on the latent space model using CVX. The x-axis is set by $C := n(p-q)^2/\log n$, and the y-axis is the empirical probability of exact inference $\mathbb{P}\{Y = Y^*\}$. The growing curves match our theoretic findings in Theorem 1.

**Experiment 2**: We validate the proposed method on a large scale latent space model. In this experiment we fix the number of entities $n$ to be 10000. It is known that CVX can be inefficient to solve large scale semidefinite programs due to the large memory cost. To test the proposed algorithm with more entities, we implement a gradient method SDP solver Yurtsever et al. (2021). We use the same model parameters as in Experiment 1, except that we fix $\|\mu\| = 1$ now, and iterate $\sigma^2$ from 0.25 to 1.25 with an interval of 0.25.

We report the average number of correctly recovered labels, and the empirical probability of exact inference $\mathbb{P}\{Y = Y^*\}$, in Figure 3. Our result suggests that even in the large scale case (10000 entities in total), exact inference of the cluster structure can be achieved efficiently through a gradient method SDP solver.

**Experiment 3**: We test the runtime scalability of the gradient method SDP solver on a large scale latent space model. We iterate $n$ from 1000 to 10000 with an interval of 1000. We use the same model parameters as in Experiment 1, except that we fix $\|\mu\| = 1$, and $\sigma^2 = 0.25$ now. We report the runtime of the gradient method SDP solver in Figure 4. The runtime can be fitted almost perfectly by a third order polynomial, suggesting the efficiency of the method.

As a comparison, when $n$ is set to 1000, CVX SDPT3 solver takes 36.26 seconds, and our gradient method solver takes only 2.50. When $n$ is set to 2000, CVX SDPT3 solver runs out of 16GB memory, and our gradient method solver succeeds and takes only 15.95 seconds.

**Experiment 4**: We test the adequacy of our method on a real-world dataset *email-Eu-core* (Leskovec & Krevl, 2014), where the equal size assumption does not hold. During each trial, we extract $n$ most connected nodes from the dataset, and solve for the cluster structure in the induced subgraph. We also compare our results with the Kernighan-Lin algorithm with random initialization for 100 iterations.

We report the number of correctly recovered labels in Figure 5. The result suggests that our method performs well compared to the Kernighan-Lin algorithm, even in cases where the equal size assumption does not hold.

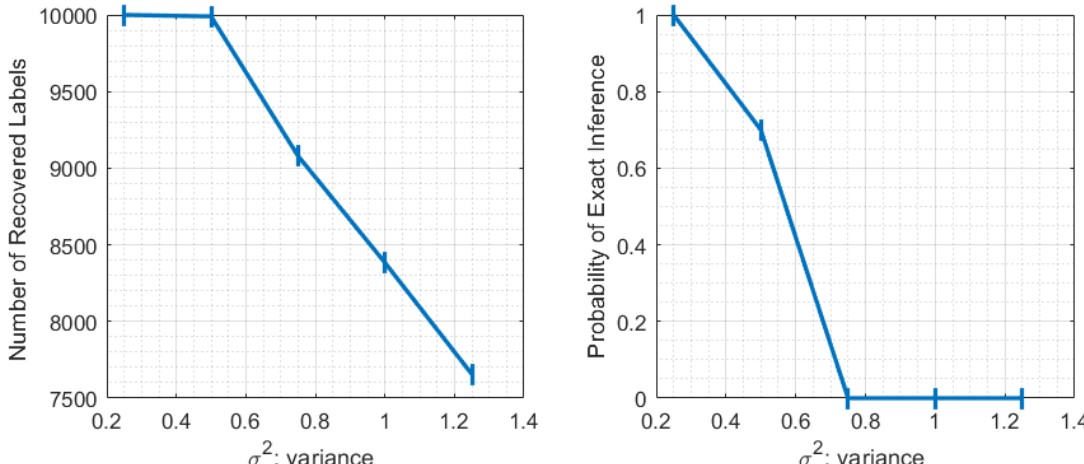

Figure 3: Simulations on the latent space model using CVX. The x-axis is the Gaussian distribution variance $\sigma^2$, and the y-axis is the number of recovered labels and the empirical probability of exact inference $\mathbb{P}\{Y = Y^*\}$, respectively. Even if $n = 10000$, as long as the variance is not too large, exact inference of the cluster structure can be achieved efficiently through the proposed method.

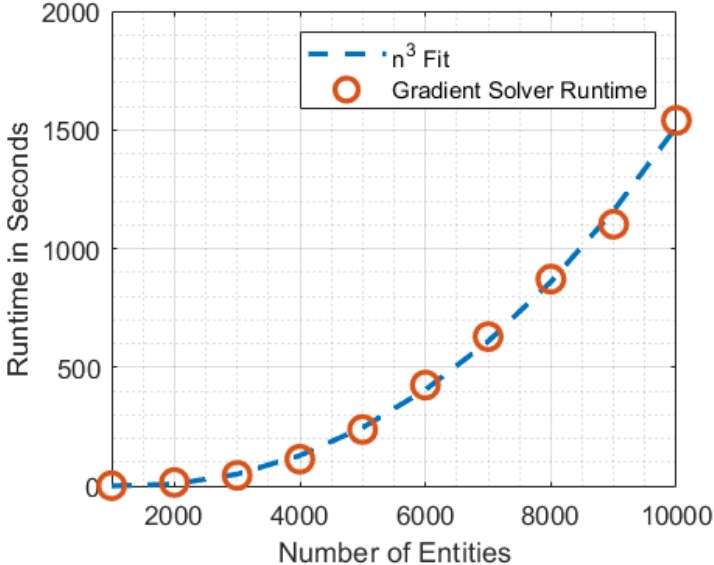

Figure 4: Runtime of the gradient method SDP solver versus the number of entities. The runtime can be fitted almost perfectly by a third order polynomial, suggesting the efficiency of the method.

## 6.1   Additional Experiments

One of our contributions in this paper, is to show that our proposed method can achieve exact inference in latent models, *without* estimating the latent vectors. In contrast, the maximum likelihood estimation (MLE) approach of recovering the cluster structure in a latent model, is to estimate all the latent vectors first, and then run clustering algorithms (e.g., K-means) using the estimated latent vectors. Here we demonstrate that our proposed method, without the need to estimate the latent vectors, can achieve even better performance than the MLE approach.

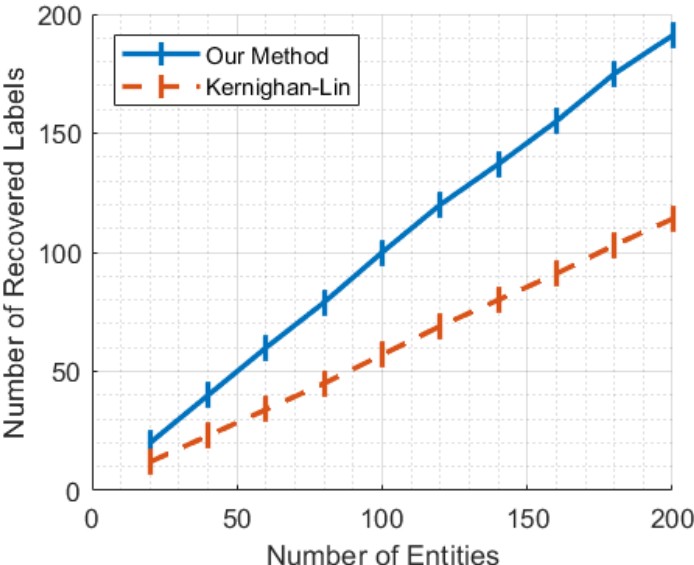

Figure 5: Simulations using the real-world dataset *email-Eu-core* (Leskovec & Krevl, 2014). In the $n = 200$ case, the two clusters have 92 and 108 entities, respectively. The result suggests that our method performs well even if the balancedness assumption does not hold.

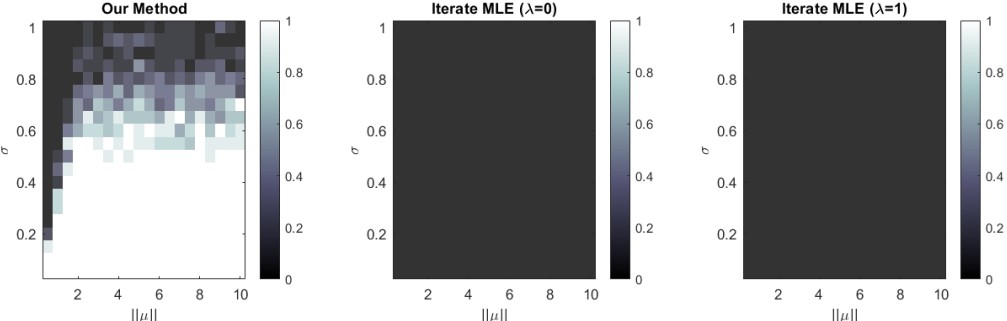

Figure 6: Probability of exact inference in latent space model. **Left:** our SDP method. **Mid:** alternate MLE with $\lambda = 0$. **Right:** alternate MLE with $\lambda = 1$. A brighter color indicates a higher probability of successful trials.

To do so, we compare the performance of our SDP method against that of an alternate MLE approach (Algorithm 1). We test both algorithms on a synthetic latent space model with two clusters. We pick $\mathcal{X} = \mathbb{R}^2$ as the latent domain. We fix the number of entities $n$ to be 100. We generate $Z^*$ by randomly assigning $n/2$ entities to one group ($z_i^* = 1$), and $n/2$ entities to the other group ($z_i^* = -1$). We generate the latent variables using Gaussian distributions, such that $P_1 = \mathcal{N}_2(\mu, \sigma^2 \mathbf{I})$, $P_2 = \mathcal{N}_2(-\mu, \sigma^2 \mathbf{I})$, where $\mathcal{N}$ denotes the Gaussian distribution. We also set $f(x, x') = \exp(-\|x - x'\|^2)$. The parameters in our simulations are $\|\mu\|$ and $\sigma$. Each entry $W_{ij}$ follows Bernoulli distribution with probability $f(x_i, x_j)$.

The motivation behind Algorithm 1 is as follows. The algorithm first initializes the latent vectors $x_1, \ldots, x_n$ randomly. Then for each latent vector $x_i$, the algorithm seeks to maximize its likelihood based on the observation of affinity $W_{ij}$'s and other latent vector $x_j$'s. Since in the latent space model $W_{ij}$ follows Bernoulli distribution with probability $f(x_i, x_j)$, the likelihood function of $x_i$ is

$$\prod_{j \neq i} f(x_i, x_j)^{W_{ij}} (1 - f(x_i, x_j))^{1 - W_{ij}},$$

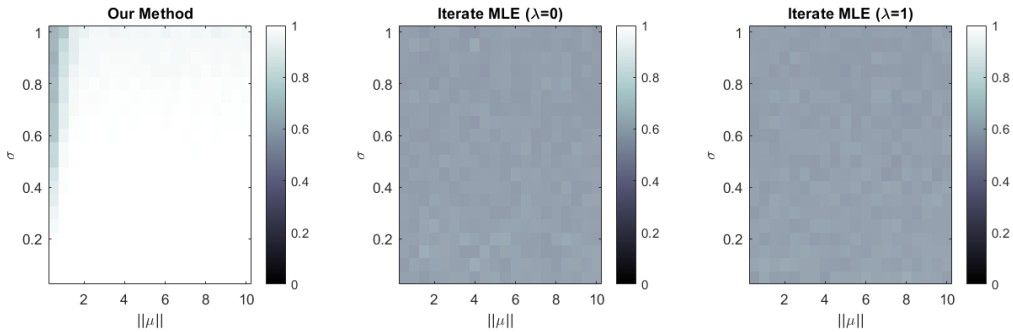

Figure 7: Percentage of recovered labels in latent space model. **Left:** our SDP method. **Mid:** alternate MLE with $\lambda = 0$. **Right:** alternate MLE with $\lambda = 1$. A brighter color indicates a higher percentage of recovered labels.

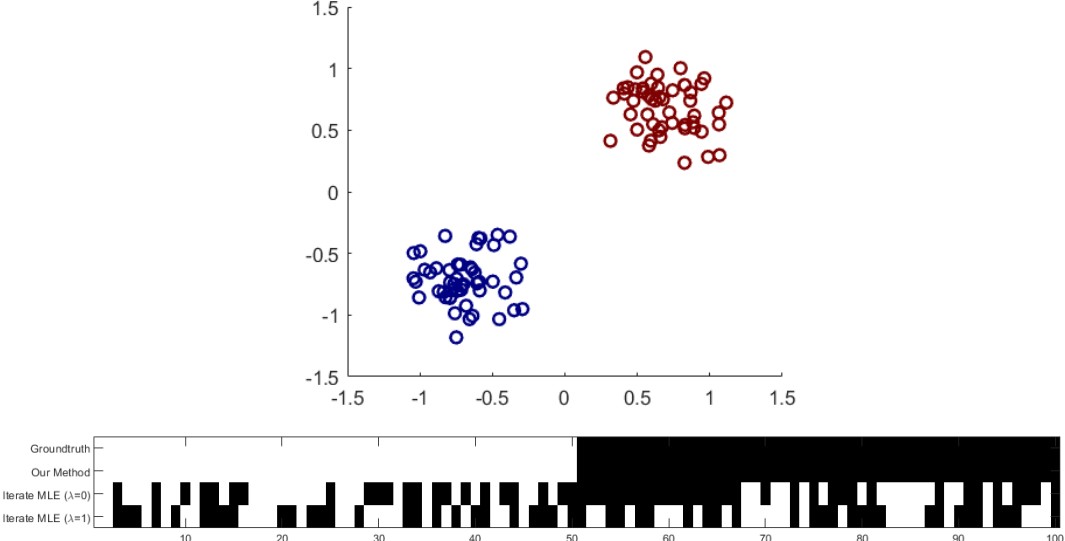

Figure 8: Latent space model with $\|\mu\| = 1, \sigma = 0.2$. **Top:** distribution of latent vectors. **Bottom:** distribution of true labels, labels predicted by our method, and labels predicted by alternate MLE.

which has a negative log likelihood of

$$\sum_{j \neq i} \left( W_{ij} \|x_i - x_j\|^2 + (W_{ij} - 1) \log \left( 1 - \exp(-\|x_i - x_j\|^2) \right) \right) .$$

Reweighting the expression above and adding an Euclidean regularization term, we obtain the following objective function

$$\frac{1}{n-1} \sum_{j \neq i} \left( W_{ij} \|x_i - x_j\|^2 + (W_{ij} - 1) \log \left( 1 - \exp(-\|x_i - x_j\|^2) \right) \right) + \frac{\lambda}{2(n-1)} \|x_i\|^2 .$$

We solve for $x_i$'s using the MATLAB built-in `fminunc` solver. After that, the algorithm uses 2-means algorithm to recover the cluster structure.

It is worth highlighting that our SDP solver does not have access to the side information in Algorithm 1. First, the cost function in the MLE solver perfectly matches the function $f$ being used in the generative

---

**Algorithm 1** Alternate Maximum Likelihood Estimation

---

**Input:** Affinity matrix $W \in \{0,1\}^{n \times n}$, latent space dimension $d$, regularization parameter $\lambda$
**Output:** Predicted label vector $\hat{z}$

  $x_1, \ldots, x_n \leftarrow \mathcal{N}_d(0, \mathbf{I})$ {random Gaussian initialization of latent vectors}
  **while** not converge **do**
    **for** $i = 1, \ldots, n$ **do**
      $x_i \leftarrow \arg\min_{x_i} \frac{1}{n-1} \sum_{j \neq i} \left( W_{ij} \|x_i - x_j\|^2 + (W_{ij} - 1) \log \left( 1 - \exp(-\|x_i - x_j\|^2) \right) \right) + \frac{\lambda}{2(n-1)} \|x_i\|^2$
    **end for**
  **end while**
  $\hat{z} \leftarrow \texttt{K-means}(\{x_1, \ldots, x_n\}, 2)$ {2-means clustering using estimated latent vectors}

---

process, which gives a bonus to the algorithm. Second, Algorithm 1 takes the true dimension of the latent space $d$ as an input, while our SDP method does not observe that at all. Thus, the proposed SDP method has much less information than the alternate MLE method.

We run each trial for 10 times. We report the empirical probability of achieving exact inference in Figure 6, and we report the average percentage of correctly recovered labels in Figure 7. The figures suggest that our method outperforms the alternate MLE approach using both metrics. To help better comparing the two methods, we visualize an instance of the latent space model with $\|\mu\|^2 = 1, \sigma = 0.2$ in Figure 8. We plot the true distribution of the latent vectors, as well as the true labels and the labels predicted by the two methods. The result suggests that even in this simple case, the alternate MLE method cannot produce accurate inference results, while our method produces perfect results.

*Remark* 8. We sum up the comparison. The alternate MLE method estimates the latent vectors, while our SDP method does not. The alternate MLE method utilizes the side information about the functions and the dimension being used in the generative process, while our SDP method does not have access to any side information at all. Nevertheless, our proposed method performs significantly better than the alternate MLE approach.

## 7 Future Work

In this paper, we considered the problem of exact inference in latent models characterized by an arbitrary latent domain $\mathcal{X}$ and a binary homophily function $f : \mathcal{X} \times \mathcal{X} \to [0, 1]$. We provided efficient algorithms, as well as theoretical guarantees for inference.

It remains an open question, if our current results could be generalized to models involving higher order interactions. In other words, is it possible to give exact inference guarantees and efficient algorithms, if $f$ takes more than two parameters? A naive approach is to use supersymmetric tensors to characterize the homophily between multiple entities. This could be a direction in the future.

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
