# OpenReview forum: "Exact Inference with Latent Variables in an Arbitrary Domain"
_TMLR — Rejected by TMLR_

### Review · Reviewer_HMyr · 2023-10-04

**Summary Of Contributions:**

This paper is mostly a theoretical work on the problem of balanced graph partitioning. From a stochastic $n \times n$ square matrix of node affinities, the problem is to partition the nodes into $k$ groups of equal size while maximizing the sum of intra-cluster affinities. The main result of the paper is Theorem 1, which shows that an SDP relaxation of the problem (polynomial time) yields an optimal solution with high probability, for a certain class of models (so-called latent models), under certain parametric conditions. The paper presents also an experimental study on simulated data to support the author's theoretical claims.

**Audience:**

Yes

**Broader Impact Concerns:**

No concern

**Claims And Evidence:**

Yes

**Requested Changes:**

Critical:

 - rename the paper to resolve the ambiguity around "exact inference with latent variables"

 - introduce the problem statement in the preliminary section

 - explain the poor performance of the "alternate MLE" baseline, or even better, compare to a stronger baselines

Strengthen the work:

 - fix all of the issues listed as Weaknesses above

**Strengths And Weaknesses:**

### Strengths

 - the presented results seem sound and novel, although I am not an expert in the field of graph partitioning nor SDP relaxation and I did not check the proofs
 - I believe the results could be of interest to some people


### Weaknesses

 - as a non-expert in this specific field, I had a hard time understanding what was the actual problem tackled in the paper. It is unfortunate that the authors broadly employ the term "inference" to refer to the specific problem of balanced graph partitioning assuming an unknown latent variable model. There is a whole field of machine learning dedicated to latent variable models, where inference can mean many other things, from learning the model parameters to computing the probability of an event or predicting the most probable answer to a query. See for example [1] or [2]. I'd strongly suggest to change the paper's title, abstract and introduction so that it better describes the paper's actual contribution, unambiguously, to a broader audience.

 - the problem definition can only be found on page 9. It should be stated clearly in the preliminaries section.

 - the assumption of balanced clustering is shown on page 3 without a warning or a word of caution. Does it restrict the results in any way? Are the presented results only valid under this assumption? Is the problem harder otherwise?

 - the term "fully observed models" seems ill-suited, as the same thing is observed (affinities $W$), only the assumptions about the data-generating process change.

 - the experiments are very synthetic and small-scale. I believe there must exist real-world instances of graph partitioning problems on which the proposed method could be tested (see for example https://chriswalshaw.co.uk/partition/).

 - the baseline "alternate MLE" method against which the proposed method is evaluated performs very poorly, basically the same level as random (figs 7, 8). I find this very questionable, either it is not a suitable baseline, or I would be suspicious of an implementation error. Why was this method chosen? Why are the results so poor?

### Detailed comments

 - p1: From the title and abstract, the focus of the paper is unclear to me. Exact Inference with Latent Variables can mean many things. Exact Inference of the probability of an event, i.e., computing $p(x)=\sum_z p(x,z)$ ? Exact inference in classification, i.e., $\arg \max_y p(y|x)$ ? Exact Inference of the latent model itself, e.g., maximum likelihood $\arg \max_p \prod_{x \in \mathcal{D}} \sum_z p(x,z)$? Also, there seems to be a mismatch between what is presented here as a general result, and what is said in Section 7 (p14) "we considered the problem of exact inference in latent models characterized by an arbitrary latent domain $X$ and a binary homophily function $f: X \times X \to [0, 1]$". Both the abstract and the title must be more precise about what the paper is really about.

 - p1: (Abbe, 2018; Ke & Honorio, 2018; Fortunato, 2010) -> I searched for "inference" in these 3 references, I could not find a clear definition of "the core task of inference". "graph partitioning" is a much less ambiguous term, less dependent on the reader's background or the context in which the word is used. If this is the focus of the paper, I would suggest renaming it "Exact Graph Partitioning with Latent Variables" which is less confusing.

 - p1: problem problem -> typo

 - p1: the the

 - p2: Section 2 Preliminaries -> the problem definition is missing. Is it recovering Z from W? Recovering Z from f and W? Recovering X and f from W? recovering X from W?

 - p3: the kernel latent variable model -> ref missing

 - p3: balance case -> balanced case?

 - p3: each cluster has the same size $m=n/k$ -> what are the implications of this assumption for your method? Is the problem harder without this assumption?

 - p3: the true cluster assignment matrix -> this is a little confusing, as there is no explicit constraint that an entity belongs to one and only one cluster. If that constraint is true, then $Z^{\star}=\{1,\dots,k\}^n$ would be less ambiguous. Also, this would be more consistent with Figure 1.

 - p3: affinities are not independent if not conditioning on the latent variables. -> this sentence is quite confusing, it took me a while to understand what was meant here. I'd suggest to actually write down the independences you talk about. $W_{i,j} \perp W_{j,k} | Z_j$ is true in the "fully observed" case, but not in the "latent" case.

 - p3: $i$, $j$ and $k$ are three entities -> inconsistent notation, $k$ was used before as the number of clusters

 - p3: fully observed models -> a definition is missing. How do you describe a fully observed model in your proposed framework? Are the entities $x_i, x_j$ observed, in addition to the affinities $W_{i,j}$ ? Are the affinity functions known?

 - p3: Definition 2 -> I read this as a tautology, latent conditional independence is independence conditioned on a latent variable. Also, the general concept of independence does not apply to an arbitrary set of variables. Do you mean pairwise conditional independence?

 - p4: Figure 1 -> I find the term "fully observed model" really not well-suited and confusing. From what I understand, the difference between the right and left models is not that in one we observe more things than in the other. In both cases, only $W$ is observed. The difference is in the choices made for modelling $p(Z|W)$. I am not sure if the term "fully observed models" is widespread in the community or if it is a proposition from the authors, but I find it a really confusing choice of words.

 - p5: $<W,ZZ^T>$ Is this a dot product of some sort? I cannot find this notation in the Section 2.1.

 - p9: we would like to infer labels given the observation of the adjacency matrix W. Also note that we do not observe the collection of latent variables X. -> This problem definition should appear much earlier in the paper, in Section 2.

 - p10: SDP solver Yurtsever et al. (2021). -> bad citation style

 - p14: even in this simple case, the alternate MLE method cannot produce accurate inference results, while our method produces perfect results. -> I find it surprising that the alternate MLE method fails so dramatically while it has access to more information, and even more so in a problem as simple as the one showcased in Figure 8. Is this alternate method a strong baseline to compare against? Why was this method chosen as a baseline, is it the state-of-the-art? Does it always perform so badly? It is very questionable to have a baseline method that does no better than random.

[1] Kevin Winner, Debora Sujono, Dan Sheldon, ICML 2017, Exact Inference for Integer Latent-Variable Models

[2] Christophe Dupuy, Francis Bach, JMLR 2017, Online but Accurate Inference for Latent Variable Models with Local Gibbs Sampling

---

> ### Author Response · Authors · 2023-11-19
>
> > the presented results seem sound and novel
> >
> > could be of interest to some people
>
> We thank the reviewer for the appreciation of our work.
>
> > the authors broadly employ the term "inference"... change the paper's title, abstract and introduction
> >
> > rename the paper
> >
> > p1: From title and abstract, the focus... is unclear to me... p14 "we considered the problem..."
> >
> > p1: I suggest renaming it "...Graph Partitioning..."
>
> We will add "community detection" (CD) in the title, abstract and introduction since our discussion is in terms of social network models. CD is a Markov random field inference problem. We believe p14 "we considered the problem..." is correct, but we can remove the paragraph.
>
> > problem definition can only be found on p9
> >
> > introduce the problem statement in the preliminary section
> >
> > p2: problem definition is missing
> >
> > p9 "we would like to infer labels Z given the observation of the adjacency matrix W. Also note that we do not observe the collection of latent variables X" -> Section 2
>
> We will do so.
>
> > assumption of balanced clustering... Is the problem harder otherwise?
> >
> > p3: each cluster... same size $m=n/k$ -> problem harder without this...?
>
> No, after few changes. If cluster sizes are $s=(s_1,...,s_k)$, then $Z^T1_n=s$. Note that $s=m1_k$ in problems (1)-(3).
>
> > term "fully observed models" seems ill-suited
> >
> > p3: definition is missing... $x_i,x_j$ observed?
> >
> > p4: Figure 1: only $W$ is observed. The difference is in... modelling $p(Z|W)$
>
> p2 states "These models are called fully observed, because there are no latent variables, and edges are generated based on the agreement of entity labels." See also Figure 1 (left).
>
> Only the random variables $W_{ij}$'s are observed in both models. The challenge in the latter comes from a more complex generative model that depends on latent variables $x_i$'s. The power of our method is that we do not need to guess the latent variables' values in order to infer the true group assignments.
>
> Recall that $z_i^*$'s are the true (constant) labels, i.e., not a random variable. We use the term "latent variable" and "fully observed" in terms of random variables. Also, since $z_i^*$'s are not random variables, one cannot define $p(Z|W)$.
>
> > experiments... very synthetic and small-scale... real-world instances
>
> Our main goal is to validate our theoretical guarantees through synthetic experiments. We tested up to $n=10000$ nodes. Vanilla solvers such as CVX scale only to small problem sizes ($n \leq 200$). We implemented an interior point method which easily scales to large problem sizes ($n \geq 10000$). There also exist efficient solvers for $n=10000000$ such as Yurtsever et al. 2021. The real-world experiment tested the effect of the balancedness assumption.
>
> Given the above, we are unsure of which theory insights would be gained by the suggestion.
>
> > baseline "alternate MLE" (Alt-MLE)... performs very poorly... Why was this method chosen?
> >
> > explain the poor performance... or even better, compare to a stronger baselines
> >
> > p14: has access to more information... is it the state-of-the-art?
>
> The power of our method is that we do not need to guess the latent variables $x_i$'s values in order to infer the true group assignments. p11 states "Here we demonstrate that
> our proposed method, without the need to estimate the latent vectors, can achieve even better performance than the MLE approach."
>
> We are not aware of any method for the problem of estimating $x_i$'s and then inferring group assignments. We came up with Alt-MLE by thinking about what would a reasonable ML practitioner would do:
> - Assume extra knowledge: $d$, $x_i \in \mathbb{R}^d$ and $f(x,x')=\exp(-\\\|x-x'\\\|^2)$
> - Maximize likelihood for $x_i$'s (see equations in p12-p13 and Algorithm 1)
>
> This leads to a highly non-convex problem, and Alt-MLE gets trapped in local minima.
>
> We avoid this by not exploring the space of latent variables $x_i$'s. In fact, we do not even know the domain $\mathcal{X}$ in $x_i \in \mathcal{X}$ or the specific $f(x,x')$. We directly solve the SDP (3).
>
> We highlight that TMLR guidelines for reviewers specifically state "it should not be used as a reason to reject work ... because it isn't achieving a new state-of-the-art on some benchmark." We kindly refer to https://jmlr.csail.mit.edu/tmlr/reviewer-guide.html
>
> > p3: no explicit constraint that an entity belongs to one and only one cluster
>
> The introduction states "every entity belongs to one of k groups."
>
> > p3: write down the independences
> >
> > p3: Def 2, pairwise conditional independence?
>
> Def 2 should be "if $v_1,v_2$,... are mutually conditional independent". Mutual independence implies pairwise independence.
>
> > p3: i, j and k are three entities -> k was... number of clusters
>
> We will use $l$ in that paragraph.
>
> > p5: $\langle W, ZZ^T \rangle$... dot product?
>
> We will add this standard notation in Section 2.1: $\langle A,B\rangle=\sum_{ij}A_{ij}B_{ij}$
>
> We will make the other proposed minor corrections.

---

### Review · Reviewer_gT2D · 2023-10-08

**Summary Of Contributions:**

This work proposes a semidefinite relaxation (SDR)-based method to address a graph clustering problem. A generative model is proposed, where latent variables are associated with the cluster membership of the nodes. A semidefinite programming (SDP)-based method is proposed to address the formulated maximum likelihood estimation problem. Properties of the SDP are studied, and the recovery of the ground-truth cluster membership was investigated. Some simulations and an experiment using an email dataset are conducted.

**Audience:**

No

**Broader Impact Concerns:**

No concerns

**Claims And Evidence:**

No

**Requested Changes:**

Please address the points raised in the "weakness" part

**Strengths And Weaknesses:**

Strength:

The manuscript studies a problem that is of broad interest. It provides a comprehensive study, including the model, formulation, optimization strategy, and characterization of key properties, such as the optimality of the SDP, the tightness of the relaxation, and the recoverability/identifiability of cluster membership. The algorithm was also tested using real data. Overall, the submission has made a decent attempt to close the loop on the problem of interest.

Weaknesses:

However, the reviewer thinks that many key components mentioned above came with unsatisfactory executions. The development of many parts is a bit disappointing. To be more specific, the reviewer has concerns about the following aspects:

**Writing, Organization, and Clarity**: The paper is, in general, not easy to follow. The title and abstract do not even mention graph clustering but only mention "latent model" and "inference." These terminologies have little context to be understood. The reader could only understand that the submission is about graph clustering by looking at the introduction.

The reviewer feels that the writing of this submission may hope to make the setting general and broad. But not giving exact examples makes following the setting quite hard. For example, the term "latent model" is quite hard to understand. The model in Fig. 1 also has little explanation. There is no mention of how X and Z^* are explicitly related, but the formulation aims to recover Z^* through the graph realization of X. The clarity of these parts needs significant improvements.

**Discussion regarding existing work is lacking**: The reviewer suspects that the proposed model is heavily related to graph Laplacian-based clustering (like those used in spectral clustering). The generative model perspective is similar to that used in the stochastic block model (SBM) and mixed-membership stochastic block model (MMSB). But such connections were never discussed. The proposed generative model always stayed at a high level yet never used any explicit examples to illustrate.

What is more concerning is that some claims are ambiguous and inaccurate, if not misleading. For example, "To the best of our knowledge, we are providing the first polynomial-time method for a generally computationally hard problem with formal guarantees." This statement seems to be really inaccurate, as there have been tons of SDP-based methods proposed to solve hard problems; see (Luo et al. 2010) for a survey. Even if the submission meant to only talk about using SDP to solve graph clustering/clustering problems, there has been a large number of literature covering efforts starting from more than one decade ago; see [R1-R7]. The relationship between this submission and the existing literature remains quite unclear. Compared to these existing methods, what is the additional knowledge or new analytical technique proposed in this work?

In the paper, there is mention of "We first consider a maximum likelihood estimation approach to recover the true assignment Z^*. The use of MLE in graph partitioning and community detection literature is customary (Bandeira, 2018; Abbe et al., 2016; Chen & Xu, 2014)." But there is no discussion regarding the additional value of this particular work.

**Motivation is unclear**: In Fig. 1, the submission argued that the generative model is different from existing methods. To be specific, the considered model has one layer of random models represented by X. However, the motivation for considering the new model was not explained. Why consider this particular model? What can this new model offer in terms of better modeling of graph data? If one hopes to associate a node with different clusters with a certain probability, then MMSB is a popular model to do so. What are the additional values compared to MMSB?

**Assumptions are not well supported**: Throughout this submission, there are many technical assumptions that may not hold in practice. Some even do not seem to have any reasonable physical meaning. For example, the paper assumes that the cluster sizes are exactly the same, i.e., m=n/k. How important is this assumption in the proofs? Will the proof still hold if the assumption does not hold? In addition, the condition (7) for attaining a unique solution of the SDP involves a variable Tau. Consequently, this condition is not a characterization of the problem structure. There is no physical meaning to this condition, nor can this condition be easily guaranteed in practice as the optimization variable changes all the time.

In the main theorem, the assumption "sum i,j(EW [Wij | X] - EWX [Wij ]) is sub-Gaussian for all Y" is also hard to parse. Y is an optimization variable that could change along the iterations. It is also hard to understand what it exactly means by "for Y." How to justify this assumption was not discussed at all. Some examples would help. For example, in the setting of Experiment 1 where f is the Gaussian kernel, can the assumption hold? These are unclear to the reviewer.

**Usefulness in Real Data is Unclear; Lacking benchmarks**: There is no comparison with any known methods/models in graph clustering or community detection. The simulations were only run with the proposed method. The real data experiment was compared with the Kernighan-Lin algorithm from the 1970s. There has been a vast volume of literature and a lot of progress since 53 years ago for graph clustering. Methods like SBM, spectral clustering, and MMSB are obvious baselines. These approaches all admit polynomial time solvers using either spectral methods, SDP, or tensor/nonnegative matrix factorization approaches; see [R1-R10]. If the proposed "latent model" based approach is not contrasted with these baselines, it is unclear if the new model is more useful than the existing models.

(Luo et al. 2010) Luo, Zhi-Quan, et al. "Semidefinite relaxation of quadratic optimization problems." IEEE Signal Processing Magazine 27.3 (2010): 20-34.

[R1] Peng, Jiming, and Yu Wei. "Approximating k-means-type clustering via semidefinite programming." SIAM journal on optimization 18.1 (2007): 186-205.

[R2] Ling, Shuyang, and Thomas Strohmer. "Certifying global optimality of graph cuts via semidefinite relaxation: A performance guarantee for spectral clustering." Foundations of Computational Mathematics 20.3 (2020): 367-421.

[R3] Boedihardjo, March, Shaofeng Deng, and Thomas Strohmer. "A performance guarantee for spectral clustering." SIAM Journal on Mathematics of Data Science 3.1 (2021): 369-387.

[R4] Yan, Yan, Chunhua Shen, and Hanzi Wang. "Efficient semidefinite spectral clustering via Lagrange duality." IEEE Transactions on image processing 23.8 (2014): 3522-3534.

[R5] Kim, Chiheon, Afonso S. Bandeira, and Michel X. Goemans. "Stochastic block model for hypergraphs: Statistical limits and a semidefinite programming approach." arXiv preprint arXiv:1807.02884 (2018).

[R6] Ricci-Tersenghi, Federico, Adel Javanmard, and Andrea Montanari. "Performance of a community detection algorithm based on semidefinite programming." Journal of Physics: Conference Series. Vol. 699. No. 1. IOP Publishing, 2016.

[R7] Abbe, Emmanuel, Afonso S. Bandeira, and Georgina Hall. "Exact recovery in the stochastic block model." IEEE Transactions on information theory 62.1 (2015): 471-487.

[R8] X. Mao, P. Sarkar, and D. Chakrabarti, “On mixed memberships and symmetric nonnegative matrix factorizations,” in International Conference on Machine Learning, 2017, pp. 2324–2333.

[R9] M. Panov, K. Slavnov, and R. Ushakov, “Consistent estimation of mixed memberships with successive projections,” International Workshop on Complex Networks and their Applications, pp. 53–64, 2017.

[R10] Anandkumar, Animashree, et al. "A tensor spectral approach to learning mixed membership community models." Conference on Learning Theory. PMLR, 2013.

---

> ### Author Response · Authors · 2023-11-19
>
> > problem of broad interest... comprehensive study... made a decent attempt to close the loop
>
> We thank the reviewer for the appreciation of our work.
>
> > The title and abstract... only mention "latent model" and "inference"
>
> We will add "community detection" in the title and abstract, since our discussion is in terms of social network models.
>
> > not giving exact examples makes following the setting quite hard.. There is no mention of how X and Z^* are explicitly related
>
> Section 2 states "Let $Z^*\in\{0,1\}^{n \times k}$ be the true cluster assignment matrix, such that $Z^*_{ij}=1$ if entity i is in cluster j, and $Z^*_{ij}=0$ otherwise. For every entity i in cluster j, nature randomly generates a latent vector $x_i\in\mathcal{X}$ from distribution $P_j$. A random observed affinity matrix $W\in[0, 1]^{n\times n}$ is generated, such that the conditional expectation fulfills $E[W_{ij}|x_i,x_j]=f(x_i,x_j)$."
>
> > similar to... stochastic block model (SBM)
> >
> > Fig. 1 also has little explanation
>
> The latent model is a strict generalization (and more challenging) than fully observed models, which includes the SBM.
>
> The introduction states "In the past decade, there have actually existed a large amount of literature on network models, and most focus on the class of fully observed models, for example... the Stochastic Block Model. These models are called "fully observed", because there are no latent variables, and edges are generated based on the agreement of entity labels." See also Figure 1 (left).
>
> Only the random variables $W_{ij}$'s are observed in both models. The challenge in latent models comes from a more complex generative model that depends on latent variables $x_i$'s. The power of our method is that we do not need to guess the latent variables' values in order to infer the true group assignments.
>
> > The proposed... model... never used any explicit examples to illustrate
> >
> > the motivation for considering the new model was not explained
>
> The summary of our contributions state "We propose the definition of the latent model class, which is highly general and subsumes several latent models from prior literature (see Table 1)." This includes, e.g., latent space model [Hoff'02], the
> exchangeable graph model [Goldenberg'10], the dot product graph (DPG) [Nickel'08], the uniform DPG [Young&Scheinerman'07], the extremal vertices model [Daudin'10].
>
> > "we are providing the first polynomial-time method for a generally computationally hard problem with formal guarantees."... The relationship between this... and the existing literature remains quite unclear
>
> The next sentence gives more context "In general, problems involving latent variables are computationally hard and nonconvex".
>
> > "The use of MLE in graph partitioning and community detection literature is customary"... there is no discussion regarding the additional value of this particular work
>
> While SDP was used before for SBM, our theoretical analysis shows that SDP can tackle a more challenging setting: latent models.
>
> > SBM are obvious baselines... using SDP
>
> While the theoretical analysis is different, the SDP formulations between SBM and ours are similar.
>
> > similar to... mixed-membership stochastic block model (MMSB)
> >
> > node with different clusters with a certain probability, then MMSB is a popular model
> >
> > MMSB are obvious baselines... using tensor/nonnegative matrix factorization
>
> The introduction states "every entity belongs to one of k groups." In our model, $P_j$ is not a scalar, $P_j$ is a distribution, e.g., in the latent space model, $P_j$ is a Gaussian distribution. See Section 4.4.
>
> Thus, we did not compare experimentally.
>
> > heavily related to... spectral clustering (SC)
> >
> > spectral clustering (SC)... are obvious baselines
>
> Our main goal is to validate our theoretical guarantees through synthetic experiments. Although we are unsure of which theory insights would be gained, we could add the SC heuristics, with either eigenvector recursive partitioning, or eigenvectors' $k$-means. This already puts SC at a disadvantage.
>
> > cluster sizes are... m=n/k... proof still hold if the assumption does not hold?
>
> Yes, with few changes. If cluster sizes are $s=(s_1,...,s_k)$, then $Z^T1_n=s$. Note that $s=m1_k$ in problems (1)-(3).
>
> > condition (7)... involves a variable... changes all the time
>
> The condition for the dual variable $\Lambda$ are not for any value of $\Lambda$ but for the optimal $\Lambda$. We set $\Lambda$ in our primal-dual construction using KKT conditions (sufficient and necessary for convex problems). Conditions on dual variables are used in prior analyses, e.g., for SBM.
>
> > assumption "... is sub-Gaussian for all Y" is also hard to parse
>
> We are sorry for the confusion caused by a typo. We will remove "for all Y" and also "for Y" in p23.
>
> Finally, we noticed that the reviewer chose "Claims And Evidence: No" as well as "Audience: No". Please let us know if there is anything else that we are missing.

---

### Review · Reviewer_D98R · 2023-11-03

**Summary Of Contributions:**

Authors study the problem of exact inference in a specific type of latent variable model for clustering problems. The paper studies the probability that a convex optimization ("SDP relaxation") approach succeeds in inferring the group/cluster assignments of the observed samples. The paper also studies the regimes where exact inference is impossible for any algorithm and the "computationally hard" regime where non-convex max. likelihood estimator succeeds but the convex relaxation fails.

**Audience:**

Yes

**Claims And Evidence:**

No

**Requested Changes:**

see above

**Strengths And Weaknesses:**

I find the techniques ("Latent conditional indepence concnetration results") used to analyze the latent model quite interesting. However, it is unclear how these technical devices fit in the main theme of the paper. If the main theme is the latent model (shown in Fig 1 right) and its computational and statistical properties, then i would not present the concentration results so prominently but rather in an appendix.

The authors should discuss in more detail how the studied latent model is related to stochastic block models for clustering problems (including the work ob Abbe that is already cited) as well as to work on machine learning over networks (see [Ref1]). Having clarified these relations would also set the stage for comparing the results of your analysis with known results on SBM inference (impossible, hard regimes) as well as bounds on estimation errors of machine learning methods (see [Ref1]).

In general, the paper could benefit from improving the clarity of presentation and use of language:

* Section 1 should have one paragraph that explains the plan/outline of the paper. I feel that one useful outline could looks as follows: Section 2: problem setting; Section 3: Exact inference via SDP; Section 4: Fundamental Limits; Section 5: Numerical Experiments  (i would not put so much focus on the LCI concentration techniques unless these are really the main result of your work)

* Section 2 should be expanded as provide a clear description of the problem setting, including the definition of "exact inference" as well as "impossible regime" and "hard regime"

* use of language can be improved e.g.,
--- "..we consider the balance case in this paper..."
--- "Additional  Analysis" is not very informative as section title
--- "We call this the hard regime because without some convex relaxation .."

---

> ### Author Response · Authors · 2023-11-18
>
> > I find the techniques ("Latent conditional indepence concnetration results") used to analyze the latent model quite interesting. However, it is unclear how these technical devices fit in the main theme of the paper.
>
> We thank the reviewer for the appreciation of our work.
>
> Section 1 states "As a byproduct of our analysis, we provide latent conditional independence (LCI) concentration inequalities, which are a key component in the analysis of latent models (see Remark 7). To the best of our knowledge, these results are novel and could be useful for many other latent model problems."
>
> Remark 7 states "The LCI inequalities are a key component in our analysis. This is because in our model, $W$ depends on $E_W [W | X]$, and $E_W [W | X]$ depends on $E_{WX} [W]$. Regular matrix concentration requires two steps: concentrating the observed random matrix $W$ around the conditional expectation $E_W [W | X]$ using matrix Bernstein inequality, then concentrating $E_W [W | X]$ around the total expectation $E_{WX} [W]$ using matrix Chebyshev inequality. This approach gives the final bound in the form of $(p - q)^2/k^2 = \Omega(n^2)$, which is much worse than the $\Omega(\log n/n)$ rate. On the other hand, the LCI inequalities can be applied in one single step in the analysis, while giving us a tighter bound."
>
> > The authors should discuss in more detail how the studied latent model is related to stochastic block models for clustering problems (including the work ob Abbe that is already cited) as well as to work on machine learning over networks (see [Ref1])...
>
> The latent model is a strict generalization (and more challenging) than fully observed models, which includes the Stochastic Block Model.
>
> The introduction states "In the past decade, there have actually existed a large amount of literature on network models, and most focus on the class of fully observed models, for example... the Stochastic Block Model. These models are called "fully observed", because there are no latent variables, and edges are generated based on the agreement of entity labels." See also Figure 1 (left).
>
> Only the random variables $W_{ij}$'s are observed in fully observed models and latent models. The challenge in the latter comes from a more complex generative model that depends on latent variables $x_i$'s. The power of our method is that we do not need to guess the latent variables' values in order to infer the true group assignments.
>
> The reviewer mentioned [Ref1] without providing a specific reference.
>
> > Section 1 should have one paragraph that explains the plan/outline of the paper...
>
> We can definitely add this right after the summary of our contributions. We agree with the renaming of the sections.
>
> > Section 2 should be expanded as provide a clear description of the problem setting, including the definition of "exact inference" as well as "impossible regime" and "hard regime"
>
> We will borrow a short description from the beginning of Sections 4, 5.1 and 5.2 into Section 2. In addition, Section 1 states "In this paper we address the problem of exact inference in latent models with arbitrary domains. More specifically, our goal is to correctly infer the group assignment of every entity in a latent model without observing the latent variables or the latent domain." and Section 2 already contains several descriptions.
>
> > "We call this the hard regime because without some convex relaxation..."
>
> We will clarify that problem (1) is combinatorial.
>
> We will make the other proposed minor corrections. Finally, we noticed that the reviewer chose "Claims And Evidence: No". Please let us know if there is anything else that we are missing.

---

### Decision · Action_Editor_DyUG · 2024-01-08

**Recommendation:** Reject

**Comment:**

The topic and contributions of this paper are likely to be of significant interest to the TMLR community, and I recommend the authors prepare a detailed revision addressing the reviewer's concerns. In particular, Reviewers gT2D and HMyr provided a detailed list of issues ranging from presentation issues to missing discussions about assumptions, related work, and motivation. Although the authors clarified some of these issues in their response, a revision was not submitted. Before accepting this paper, a major revision is needed to clarify these concerns in the paper itself.

I encourage the authors to carefully incorporate these discussions into a major revision and re-submit to TMLR.

**Audience:**

The topic is clearly relevant to a majority of the TMLR audience.

**Claims And Evidence:**

Due to presentation issues raised by several reviewers, the validity of the claims are difficult to verify. The authors did not submit a revision addressing these concerns.

**Resubmission Of Major Revision:**

The authors may consider submitting a major revision at a later time.